# Stereomicroscopic Aspects of Non-Carious Cervical Lesions

**DOI:** 10.3390/diagnostics13152590

**Published:** 2023-08-03

**Authors:** Andreea Stănuşi, Adrian Ştefan Stănuşi, Oana Gîngu, Veronica Mercuţ, Eugen Osiac

**Affiliations:** 1Department of Prosthetic Dentistry, University of Medicine and Pharmacy of Craiova, 200349 Craiova, Romania; 2University of Medicine and Pharmacy of Craiova, 200349 Craiova, Romania; adrian.stefan.stanusi@gmail.com; 3Department of Engineering and Management of Technological Systems, Faculty of Mechanics, University of Craiova, 200585 Craiova, Romania; oana.gingu@edu.ucv.ro; 4Department of Biophysics, University of Medicine and Pharmacy of Craiova, 200349 Craiova, Romania; eugen.osiac@umfcv.ro

**Keywords:** non-carious cervical lesions, stereomicroscope, wedge-shaped

## Abstract

Non-carious cervical lesions (NCCLs) represent a form of tooth wear, characterized by the irreversible loss of dental hard tissues at the enamel–cement junction, without the involvement of caries and dental trauma. The aim of this study was to highlight the morphological elements of NCCLs via their stereomicroscopic examination and to confirm the role of this examination in the diagnosis of early lesions. In addition, the association between the morphological aspects identified during the stereomicroscopic examination of NCCLs and their etiological factors was determined. For this study, extracted teeth with NCCLs were examined with a stereomicroscope. The morphological aspects of NCCLs were evaluated at magnifications up to 75×. In wedge-shaped NCCLs, the stereomicroscopic examination allowed the identification and measurement of scratches, furrows and cracks. In saucer-shaped NCCLs, the stereomicroscopic examination highlighted the smooth appearance of the walls. The presented study highlighted the role of stereomicroscopic examination in the assessment of NCCL morphology and in their early diagnosis. The study confirmed, in particular, the role of occlusal overloads and tooth brushing in determining the morphology of NCCLs.

## 1. Introduction

Non-carious cervical lesions (NCCLs) represent a form of tooth wear, characterized by the irreversible loss of dental hard tissues at the enamel–cement junction, without the involvement of caries and dental trauma [1,2].

The prevalence of these lesions varies from less than 10% to over 90% [3], with an estimated global average of 46.7% that increases with age [4]. Several factors are considered in the etiology of these lesions, but the mechanisms involved in their etiology are biocorrosion (erosion), friction (abrasion) and occlusal stress (abfraction) [2,5]. The successful prevention and management of NCCLs require an understanding of etiological and risk factors, including how they interact over time.

Monitoring NCCLs is a treatment option in early lesions and should be based on the progression of the lesions and how they compromise the vitality, function and aesthetics of teeth [6,7]. Other treatment options are techniques to reduce dentin hypersensitivity and techniques to restore the lost tissues using restoration, possibly in combination with a surgical root-covering procedure [1,6,8,9]. 

Patients with NCCLs can benefit from alternative treatment options. Natural polymers and chitosan-based composite materials can be used to restore the lost dentine and enamel. These materials mimic the biological and mechanical properties of hard dental tissues [10]. Furthermore, patients can use natural toothpastes that improve teeth hypersensitivity and other oral diseases and conditions [11].

Restorations should be applied in the final stage of the treatment plan for NCCLs. In some cases, they are a necessity due to teeth sensitivity, physiognomic disturbances and even dental fractures in an advanced stage [7,12,13]. The marginal degradation of restorations is frequently observed over time [9] so that they do not last more than 5 years [14], with occlusal overloads having an essential role in the progression of NCCLs [6,15,16].

In this context, understanding the etiopathogenic mechanisms is essential for the management of these lesions. The clinical appearance of NCCL scan vary depending on the type and severity of the etiological factors involved [9]. The shape and size of NCCLs can provide significant data about the mechanisms involved in the genesis of these lesions and their evolution. However, a number of aspects have not been considered in studies that have focused on the relationship between lesion progression and etiological factors [7].

An evaluation of NCCL morphology in the dental office is frequently performed via direct clinical examination or by using magnification systems such as loupes and dental operating microscopes.

The clinical examination of NCCLs, frequently performed by the dentist, reveals aspects of macromorphology, such as shape, depth and location. Histological and microscopic examinations of NCCLs reveal aspects of micromorphology, such as the surface texture and the presence/absence of sclerotic dentine [17]. Also, for the detection of any surface defect and the study of the mechanisms leading to its formation, on either the root or cement, fractal dimension analysis is a useful tool [18,19].

In the specialized literature, there are studies on the morphology, etiology and evolution of NCCLs using various methods of investigation, among which we mention the following: the optical coherence tomography method (OCT) [20,21,22], micro-computed tomography [23,24], and electron microscope examination [25,26,27,28]. 

Among these methods, the OCT examination method has proven its usefulness in the diagnosis of early lesions, by identifying changes in hard dental tissues in the form of demineralization, cracks and substance loss [29,30]. Using the OCT examination method, 2D images of changes in the dental cervical hard tissues can be obtained in a non-invasive manner, without X-ray irradiation.

The examination of NCCLs using a stereomicroscope has not been mentioned so far in studies in the specialized literature. The stereomicroscope is a device with major industrial and medical applications. It has a magnification higher than the operating dental microscope, which allows it to identify cracks and measure small defects on the surfaces of the objects examined. In addition, 2D images and 3D reconstructions of the examined objects can be obtained [31].

The aim of this study was to highlight the morphological elements of NCCLs via their stereomicroscopic examination and to confirm the role of this examination in the diagnosis of early lesions. In addition, the association between the morphological aspects identified during the stereomicroscopic examination of NCCLs and their etiological factors was determined.

## 2. Materials and Methods

### 2.1. Sampling and Preparation of Teeth

The present study is a continuation of research carried out on 112 teeth, extracted from 56 patients aged 30 to 65 years, and treated at the Dental Prosthetics Clinic of the Faculty of Dentistry, the University of Medicine and Pharmacy of Craiova [29]. In that study, the teeth were examined macroscopically and using OCT on the axial surfaces, in the cervical areas. In total, 34 NCCLs were identified via macroscopic examination and OCT in 27 teeth.

This study was carried out on the 27 teeth with NCCLs obtained from the previously mentioned study. The teeth selected for this study were 13 incisors and 14 premolars. Patients provided written informed consent for the dental treatment recommended and for inclusion in the presented study. This study was approved by the Ethics Committee of the University of Medicine and Pharmacy of Craiova, Romania (no. 55/16.02.2023).

The inclusion criteria were as follows: (1) patient provided written consent to participate in the present study; (2) teeth had severe periodontal or dental diseases with no indication of conservative treatment; (3) teeth had NCCLs identified via macroscopic and/or OCT examination; and (4) patient had partial edentulism. 

The exclusion criteria were as follows: (1) prosthetic treatment; (2) endodontic treatment; (3) filling in cervical area; and (4) caries in cervical area. 

The teeth included in this study were extracted atraumatically, and then disinfected in 10% peroxide solution for 10 min. All teeth were cleaned, ultrasonically scaled and curetted. The teeth were kept in 0.9% NaCl until the stereomicroscopic examination was carried out to avoid dehydration [32,33]. All the procedures described were performed by 2 dentists, a coordinator and an operator, who are specialists in the field of dental wear, respecting the described working protocol for each tooth.

### 2.2. Stereomicroscopic Examination of NCCL

To carry out this study, the NIKON SMZ 745T stereomicroscope (Nikon Corporation, Tokyo, Japan) was used at a maximum magnification of 75× and at a working distance of 115 mm (Figure 1). The device was used to examine dental surfaces with NCCLs and to acquire their 2D images, stored as TIFF, JPG and PNG files. The acquisition of 2D microscopic images, their storage and subsequent processing were carried out using NIS-A-AMEAS software (version 2021), the specific software platform for this imaging system. The successive acquisition of 2D images and their recomposition in order to obtain a 3D image of the examined dental surfaces were carried out with the help of NIS-A-EDF software [34]. 

In order to position the teeth for the stereomicroscopic examination, each tooth was fixed in silicone of increased consistency (Zetaplus L Intro KIT, Zhermack, Badia Polesine, Italy), leaving only the dental surface with the NCCL exposed for the examination. The stereomicroscopic examination was performed by a specialist of Engineering and Management of Technological Systems and a specialist of Biophysics. The interpretation and analysis of the obtained images were carried out by 2 specialists in the field of dental wear.

## 3. Results

In total, 27 teeth with NCCL were examined using the NIKON SMZ 745T stereomicroscope, at degrees of magnification of 10.05×, 15×, 30×, 45×, 60× and 75×. For each tooth, only one NCCL was examined, resulting in 27 NCCLs being examined with the stereomicroscope, from the total of 34 NCCLs identified in the previous study [29]. For stereomicroscopic examination, we selected all the NCCLs located on the buccal surfaces of teeth (21 lesions in 21 teeth). For the other six teeth without NCCLs on the buccal surfaces, the existing NCCLs on the oral and proximal surfaces were examined using the stereomicroscope.

First, the teeth were examined using the stereomicroscope in order to highlight the macromorphological elements of NCCLs using lower magnification levels. From the total number of teeth examined, 11 teeth (40.74%) came from the maxillary arch (3 upper incisors and 8 upper premolars), and 16 teeth (59.26%) came from the mandibular arch (10 lower incisors and 6 lower premolars) (Figure 2a). 

Depending on shape, the lesions were divided into wedge-shaped NCCLs (16 lesions, representing 59% of the total NCCLs) and saucer-shaped NCCLs (10 lesions, representing 37% of the total NCCLs). A single lesion was identified (meaning 4% of the total NCCLs) with a mixed-shaped/irregular shape (Figure 2b).

To measure the depth of the NCCLs, the teeth were positioned in lateral norm and the NIS-A-AMEAS program (Figure 3) was used to draw a line parallel to the long axis of the tooth, at the junction of the coronal and gingival edges of the lesion (the yellow line in the Figure 3), and for the line perpendicular to it, to the deepest point of the lesion (red line in the Figure 3). Of the 27 NCCLs examined, 18 lesions (67%) had a depth of less than 500 µm, 3 lesions (11%) had a depth between 500 and 1000 µm, 2 lesions (7%) had a depth between 1000 and 1500 µm, and 4 lesions (15%) had a depth between 1500 and 2000 µm (Figure 2c).

Depending on the location of the NCCL and the damage to the dental hard tissues inflicted, out of the 27 lesions examined, 11 lesions (41%) affected the crown and the root of the teeth, 14 lesions (52%) affected only the root of the teeth, and 2 lesions (7%) had affected only the crown of the teeth (Figure 2d), resulting in the dental roots being more frequently affected by NCCLs.

To examine the micromorphological elements of NCCLs, the stereomicroscope was used at higher magnifications. Evaluating the texture of the NCCL surface, numerous cracks, scratches and furrows were highlighted in the case of wedge-shaped NCCLs (Figure 4).

For the identified furrows, images were taken and stored using different degrees of magnification. The NIS-A-AMEAS program enabled the measurement of the width of these furrows (Figure 5).

Also, in several wedge-shaped NCCLs, a crack was observed in the deepest area of the lesion, at the level of the internal angle, as the red arrow indicates in Figure 6.

In saucer-shaped NCCLs, these defects of the dental hard tissues were not identified. The walls of these lesions were smooth and shiny (Figure 7).

Another element of micromorphology that was evaluated in the stereomicroscopic examination of NCCLs was the quality of exposed dentin, according to Swift’s criteria [35] (Figure 8). From the total 27 NCCLs examined, 18 lesions (78%) presented sclerotic dentin score 1, 3 lesions (13%) presented a sclerotic dentin score of 2, 1 lesion (4%) presented a sclerotic dentin score of 3 and 5 lesions (5%) presented a sclerotic dentin score of 4 (Figure 9).

## 4. Discussion

NCCLs are a form of cervical dental wear that have attracted the attention of specialists over time due to their complex etiology, varied clinical signs and the symptoms and failures of restorative treatment [36,37,38].

The terminology of this condition has seen a series of changes since they were first described by Dr. Hunter in 1778 [39]. The term most frequently used to define the non-carious loss of dental cervical hard tissues was that of “abfraction”. This term was first introduced by Grippo in 1991 [40], knowing that the clinical forms of dental wear are attrition, abrasion, erosion and abfraction.

Later, in 2004, Grippo stated that the three mechanisms involved in the occurrence of dental wear (friction through abrasion/attrition, biocorrosion/erosion and stress/abfraction) usually act simultaneously, not individually, and proposed a new terminology. For wear in the dental cervical area, Grippo proposed the use of the following terms: abrasion–abfraction, corrosion–abfraction, abrasion–corrosion, biocorrosion–abfraction [41].

Recently, in 2019, IADR specialists recommended that the term “abfraction” should no longer be used to describe these lesions, as a result of multiple studies that demonstrated the involvement of several etiological factors, not just stress (abfraction), in the non-carious loss of hard dental tissues in the cervical area [42].

NCCLs have a complex clinical picture and it is important to evaluate the characteristics of each type of lesion in order to correctly identify the responsible etiological mechanisms and to choose the optimal restorative treatment [3]. Grippo and Soares proposed the systematization of the examination of these lesions via the characterization of the elements of macro- and micromorphology in order to help the clinician in the management of this dental condition [17].

The examination of the morphological elements of NCCLs has been carried out over time via various investigation methods, with optical coherence tomography and scanning electron microscopy being noted for their advantages. However, the devices used in these methods have a limited availability in research centers in the field of dental medicine and in current dental practice.

Researchers and practitioners can also use light-induced fluorescence technology [43,44], spectrophotometric analysis [45] and the developmental enamel defects index (DDE index) [46] to identify and assess tooth damage in the cervical area.

Since 1970, dentists have used magnification systems (loupes, dental operating microscope) for the diagnosis and treatment of dental diseases. These systems can have reduced (2×–8×), medium (8×–16×) and increased (16×–25×) magnification levels [47]. A disadvantage of using magnification systems such as loupes and dental operating microscopes is represented by the reduced mobility of the operator, who has to maintain a fixed position in relation to the operating field, which must also remain fixed. Users of the dental operating microscope are often affected by fatigue and discomfort in the neck and back muscles [48].

Unlike the standard dental operating microscope, a digital stereomicroscope does not force the operator to look through the eyepieces in a fixed position, but enables the image to be viewed on a large-screen monitor [48]. The first feasible stereomicroscope was invented in 1892 and commercialized in 1896 by the Zeiss AG company in Jena, Germany [49]. The stereomicroscope was introduced as a medical device in 2005, being successfully used in ophthalmic surgery, vascular surgery, otorhinolaryngology surgery, surgical education and in dentistry [48]. In the field of dental medicine, the stereomicroscope has proven to be a useful tool for examining the external morphology of teeth [50].

Stereomicroscopic examination with a low degree of magnification enables the evaluation of the morphological elements of NCCLs related to shape, depth and location.

Regarding the shape of the lesions, the NCCLs examined were either wedge-shaped, or saucer-shaped. The stereo-microscopic examination allowed, through the use of increased degrees of magnification, the establishment of the shape, even for superficial, early NCCLs, which could not have been identified via direct clinical examination.

The form of the NCCLs is determined by the predominance of one of the three main etiological mechanisms. The establishment of the shape of the lesions thus becomes an important step in therapeutic management [51,52]. The wedge-shape of NCCLs was considered an indicator of abfraction/occlusal stress is the main etiological agent [53,54]. The saucer-shape of NCCLs was considered to be determined by biocorrosion/erosion [52]. The frequent identification and examination of wedge-shaped NCCLs, compared to saucer-shaped NCCLs, was also reported in other studies [17,55,56,57,58,59,60]. This fact draws attention to the role of occlusal overloads and the need to apply occlusal therapy in NCCL treatment.

Regarding the depth of NCCLs, the stereomicroscopic examination performed in this study enabled the precise measurement of the loss of dental hard tissues. It was observed that more than half of the identified NCCLs had a depth of less than 500 µm. These incipient, superficial lesions could not have been identified via direct clinical examination, and their depth measurement could not have been accurately performed via direct clinical examination or by using the usual magnification systems.

These superficial lesions are usually associated with dentinal hypersensitivity, characterized by short and sharp pains upon the application of thermal, chemical or tactile stimuli, apparently without being attributed to any other dental pathology [61,62]. The stereomicroscopic examination carried out in this study demonstrated the existence of a small loss of substance in the dental cervical area, which exposes the dentinal tubules to the action of external stimuli with the appearance of pain. The identification of substance losses smaller than 500 µm in the dental cervical area assists the practitioner in the therapeutic management of dentine hypersensitivity. Desensitizing treatment is indicated for dentin hypersensitivity, regardless of its etiology [13,61,62]. However, if dentin hypersensitivity is a precursor of NCCLs, the initial loss of hard dental tissue indicates to the dentist that more than simply addressing the pain should be undertaken. For lesions smaller than 1 mm, if the pain is not reduced after the desensitizing treatment is performed, some specialists recommend applying a restoration [63].

The stereomicroscopic examination using increased degrees of magnification enabled the surface texture, as well as the exposed dentin, to be evaluated.

Thus, when examining the surface texture, changes were identified in the dental hard tissues depending on the main etiological mechanism. These changes were either in the form of cracks, scratches and furrows, or in the form of smooth, shiny surfaces. The deep and parallel grooves, known as lines of progression and scratch marks, draw attention to the intense activity of mechanical etiological factors in the development of NCCLs, and the smooth, shiny surface draws attention to the action of erosive factors in the development of NCCLs [3,17,28,54,64].

In some of the wedge-shaped NCCLs examined, cracks on the walls of the lesions at the internal angle, but also cracks in the hard dental tissues surrounding the lesion (enamel and cementum), were observed. These cracks mark the involvement of occlusal stress in the development of wedge-shaped lesions. The importance of occlusal stress in the development of cracks in the dental cervical area is a subject addressed by several researchers via various working methods, including the finite element method [65,66,67,68,69] and optical coherence tomography [20,21,30]. All patients included in the study had partial edentulism; therefore, all present teeth, including the ones examined in this study, were subjected to abnormal forces during the function of the dento-maxillary apparatus. The identification of cracks at the level of the internal angle of the NCCLs, in the deepest area, demonstrated that that area is the most susceptible to progressive fatigue failure, which further advances the lesion [3,70]. The examination of these cracks using the stereomicroscope provided images of superior quality, at a magnification of up to 75×, allowing, at the same time, the precise measurement of the defects thus identified. 

It is important to identify these cracks as they highlight the role of excessive loading forces caused by clenching, bruxism and temporomandibular disorder. The associations among bruxism, imbalanced occlusion and excessive occlusal forces were reported in a study published in 2023 [71]. Therefore, when NCCLs and cracks are found, the dentist will pay more attention to occlusal therapy as part of the treatment plan, and will select a restorative material with mechanical properties suitable for that specific case [5,72,73].

Also, the presence of scratches and furrows, was observed in some of the wedge-shaped NCCLs examined, which draws attention to the role of friction in the non-carious loss of dental hard tissues. The parallel disposition of the furrows, in a mesio-distal direction, was noted. This aspect has also been observed by other researchers and correlated with the tooth brushing technique, brushing force, the direction of movement of the toothbrush on the tooth surface during brushing and the toothpaste used [28,54,64,74,75,76]. Additionally, in the presented study, the stereomicroscopic examination enabled the width of these furrows to be measured. It was noted that the width of these furrows has a value close to the diameter of the filaments of medium toothbrushes (0.23–0.29 mm), as they were classified by Hughes in 2016 [77]. A decrease in the recorded values was observed as it approached the internal angle of the lesion.

The stereomicroscopic examination of NCCLs allowed the exposed dentin at an increased magnification be examined, which highlighted its clinical appearance in terms of its color and area. The degrees of magnification used in this study enabled the operators to establish the score of the sclerotic dentin. Determining the sclerotic dentin score in NCCLs enables the success of the future restorative treatment to be indicated. In the case of sclerotic dentin, the greater the score, the greater the chance that the restorative treatment using a composite resin filling will fail [3,78,79]. Examining the appearance of exposed dentine via NCCLs at an increased magnification provides the practitioner with information regarding the extent of the sclerotic process; thus, the practitioner can adopt an optimal therapeutic approach. Depending on the score of the sclerotic dentin, the practitioner can carry out an additional conditioning of the dentin or prepare it and remove the hypermineralized tissues [36,78,80].

The present study draws attention to the advantages of the stereomicroscopic examination of NCCLs by highlighting some morphological elements that can be directly correlated with the etiological mechanisms. The novelty of the present study arises from the method used to investigate NCCLs; to our knowledge, there is no other study in the literature that uses a stereomicroscope for this purpose. This study provided quality images, at a high magnification, of cracks and furrows that mark the involvement of stress and friction in the development of NCCLs.

The weak points of this study are represented by the small number of teeth examined and the lack of similar studies in the literature with which the results can be compared. Also, the study does not report information about the gender of the patients, their diet nor oral hygiene habits.

## 5. Conclusions

The presented study highlighted the role of stereomicroscopic examination in the assessment of NCCL morphology and in their early diagnosis.

The stereomicroscopic examination enabled changes in the dental hard tissues affected by NCCLs, such as scratches, furrows and cracks, to be identified.

The presented study confirmed, in particular, the role of occlusal overloads and tooth brushing in determining the morphology of NCCLs.

Through the stereomicroscopic examination of wedge-shaped NCCLs, high-quality images of furrows caused by toothbrush abrasion were obtained.

The study draws attention to the need to use a system with an increased degree of magnification in the current practice of dentistry for the early diagnosis of substance loss.

## Figures and Tables

**Figure 1 diagnostics-13-02590-f001:**
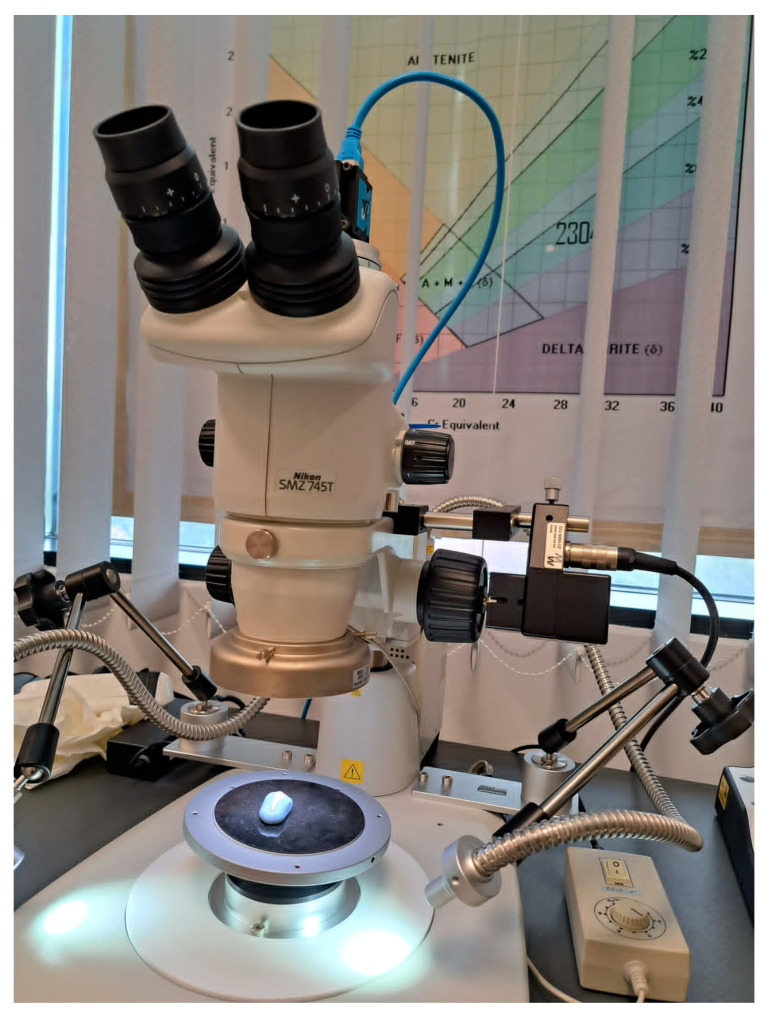
Positioning of teeth for stereomicroscopic examination.

**Figure 2 diagnostics-13-02590-f002:**
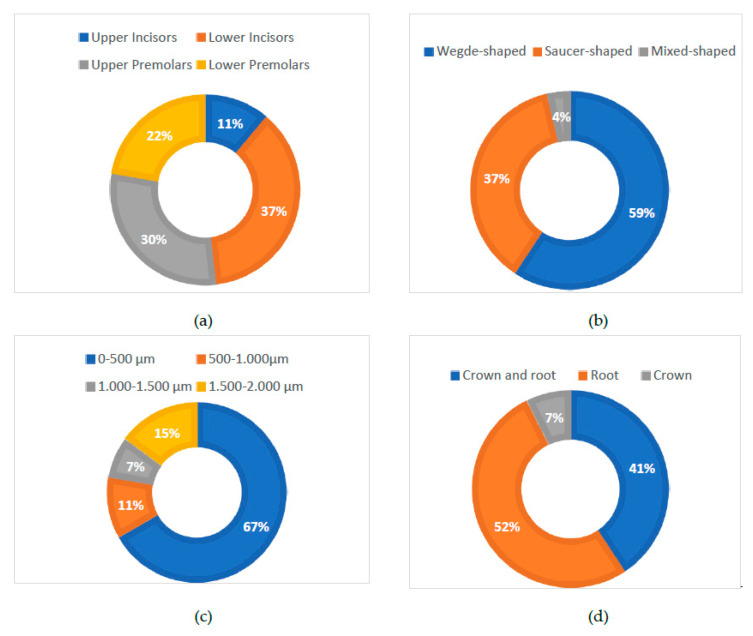
NCCL distribution: (**a**) NCCL distribution by tooth type; (**b**) NCCL distribution by form; (**c**) NCCL distribution by depth; (**d**) NCCL distribution by location.

**Figure 3 diagnostics-13-02590-f003:**
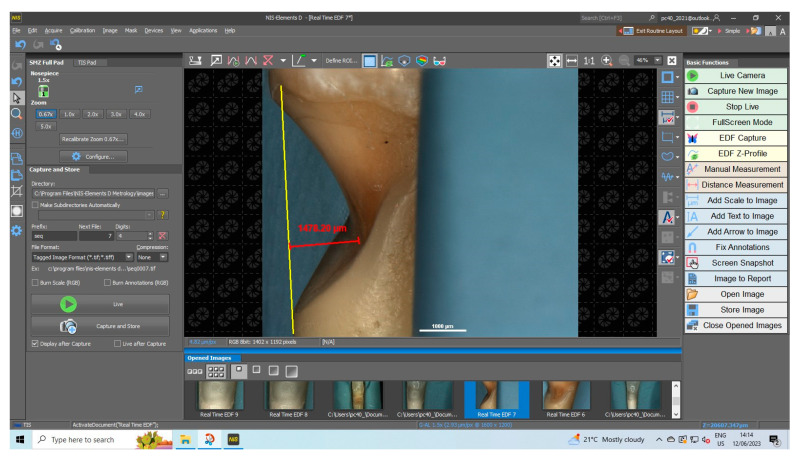
NIS-A-AMEAS program interface during NCCL depth measurement.

**Figure 4 diagnostics-13-02590-f004:**
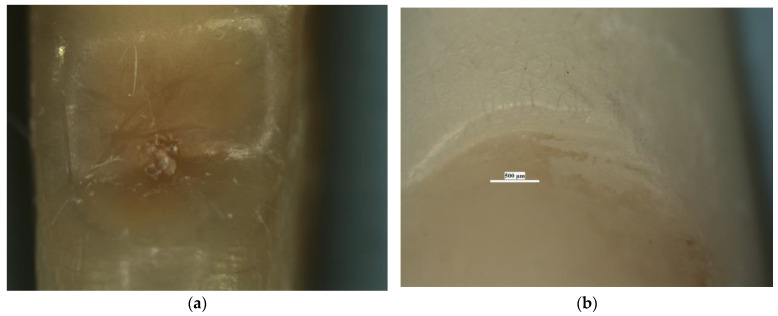
Stereomicroscopic examination of NCCLs at 30× magnification: (**a**) scratches at NCCLs of a lower incisor; (**b**) furrows at NCCLs of upper premolar; (**c**) furrows at NCCLs of lower incisor; (**d**) cracks at NCCLs of upper incisor.

**Figure 5 diagnostics-13-02590-f005:**
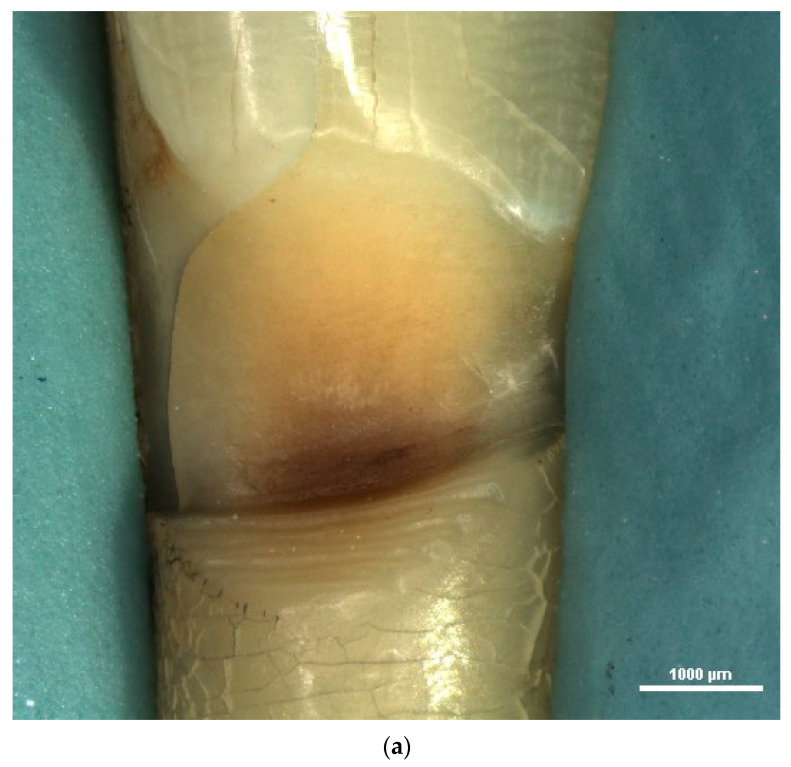
Stereo microscopic examination of a lower incisor with a wedge-shaped NCCL: (**a**) 15× magnification; (**b**) 45× magnification: the blue lines mark the measurement of the width of the furrows; (**c**) 60× magnification.

**Figure 6 diagnostics-13-02590-f006:**
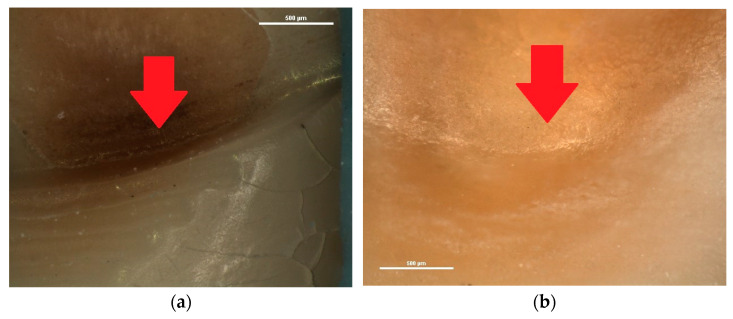
Stereomicroscopic examination of wedge-shaped NCCLs—red arrows indicate cracks: (**a**) the crack at the internal angle of the NCCL observed at 60× magnification; (**b**) the crack at the internal angle of the NCCL observed at 45× magnification.

**Figure 7 diagnostics-13-02590-f007:**
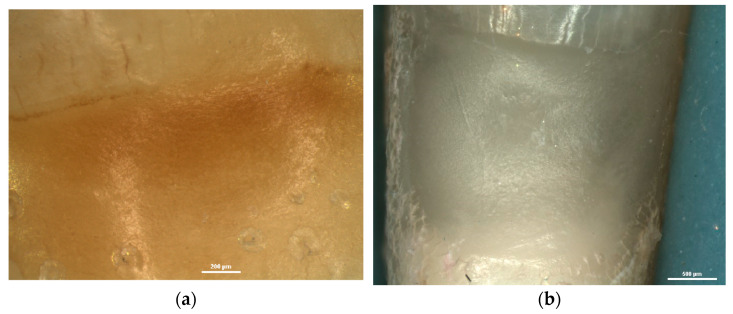
Stereomicroscopic examination of saucer-shaped NCCL—the smooth appearance of the walls: (**a**) NCCL of lower premolar seen at 45× magnification; (**b**) NCCL of lower incisor seen at 30× magnification.

**Figure 8 diagnostics-13-02590-f008:**
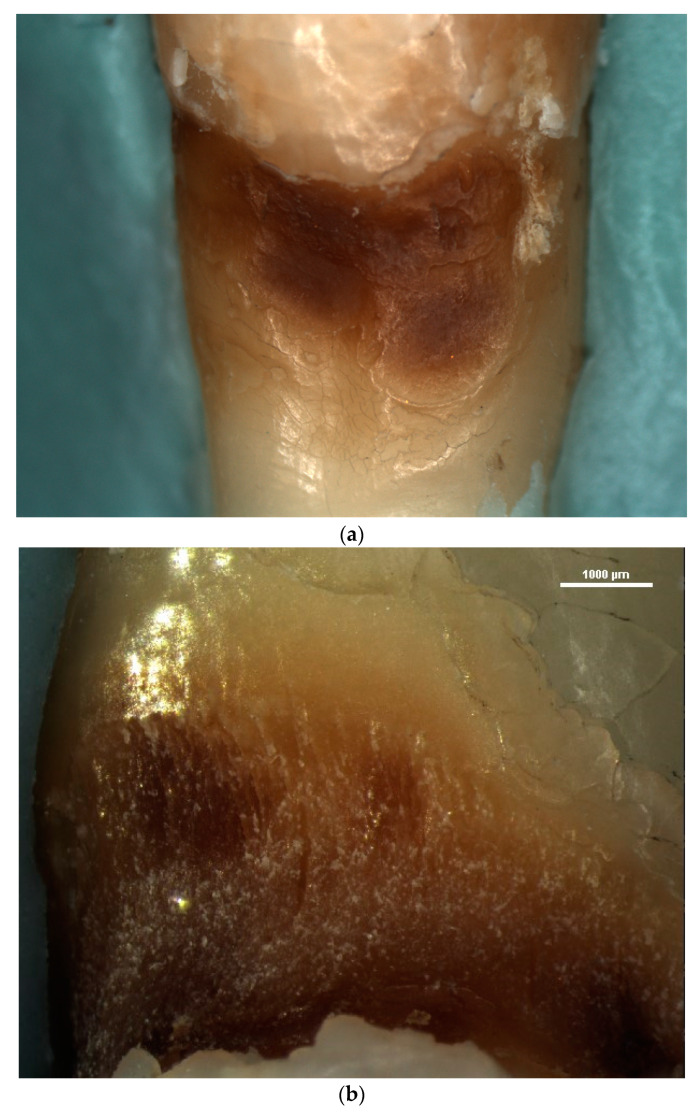
Appearance of sclerotic dentin of NCCLs: (**a**) sclerotic dentin score of 4 observed at 15× magnification; (**b**) sclerotic dentine score of 4 observed at 30× magnification; (**c**) sclerotic dentine score of 2 observed at 15× magnification.

**Figure 9 diagnostics-13-02590-f009:**
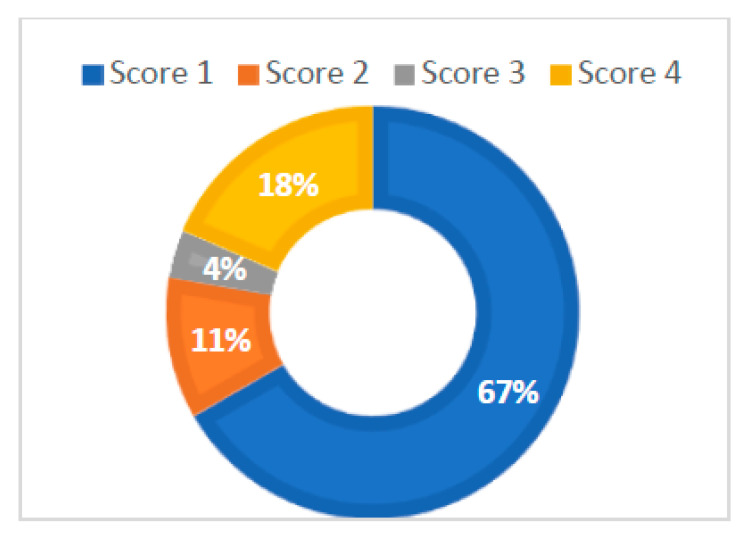
Proportion of teeth with NCCLs according to score of exposed sclerotic dentine.

## Data Availability

The authors declare that the data of this research are available from the corresponding authors upon reasonable request.

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
