# Peer review of "Stereomicroscopic Aspects of Non-Carious Cervical Lesions"

_diagnostics, 2023, doi:10.3390/diagnostics13152590_

Round 1

Reviewer 1 Report

Dear authors Please read the attached comments file.  Please appreciate that sclerotic dentin is a normal physiological response to caries or wear and is not harmful. 

Acceptable if somewhat verbose (too wordy). 

Author Response

Thank you for your review. 

Response to Reviewer 1 Comments

Thank you for the evaluation and for the recommendations!

 Point 1. Line 43 Restorations should be the last treatment option for NCCL. Why do the authors think this is the case? Surely, in some cases this is the first & most appropriate treatment option.

Response 1: Thank you for your comment. When we mentioned restorations as the last treatment option for NCCL, we were referring to restorations as being the last step in the treatment plan for NCCL. We have addressed the issue and corrected the phrase.

Point 2. Line 102 All the procedures described were done by 2 dentists, a coordinator and an operator Images were the only outcome. There is no obvious reason why 2 dentists were involved. What was the role of the coordinator?

Response 2: Thank you for your question. We considered useful the involvement of 2 dentists for the procedures described, which included the extraction, cleaning and storage of the teeth. Given the experience of the coordinator in the field of tooth wear, her role was to supervise and make sure that all the described procedures were done correctly. When the operator was working, the coordinator made sure that she did all the steps, both medical and legal. Furthermore, her experience in dental medicine and tooth wear research allowed her to assist the operator in all the difficulties she encountered.

Point 3. Line 146 Fig 3 shows the line perpendicular to the yellow line, as the deepest point of the lesion (red line) but the red line does not appear to end at the deepest point of the NCCL.

Response 3: Thank you for your comment. The end of the red line was positioned in the deepest point inside the NCCL. This point was located when examining the lesion from all angles.

Point 4. Fig 5b purports to show the width of furrows but the bracketed blue lines are lined up with what appear to be enamel ridges and not the furrows. Can the authors please check that the alignment is correct.

Response 4: Thank you for your comment. We have checked and the alignment of the lines is correct. We have measured the width of the furrows at one end, which is placed in the surrounding hard tissues. We considered that the measurement done at the end of the furrows will be more precise given the flatness of the region of interest. If we were to measure their width in the middle of the lesion, we would be facing a curved surface in more than 1 plane.

Point 5. Lines 268-270 The identification of substance losses smaller than 500 μm in the dental cervical area assists the practitioner in the therapeutic management of dentine hypersensitivity. If a patient complains of hypersensitivity, desensitizing treatment is indicated irrespective of lesion depth. Why the authors believe knowing that the lesion depth is <500 μm helps management is unclear.

Response 5: Thank you for your comment. We have addressed the issue and added more information in the revised manuscript.

Point 6. Lines 287-289 The identification of cracks at the level of the internal angle of the NCCL, in the deepest area, demonstrated that area is a stress riser and it is the most susceptible to progressive fatigue failure that further advances the lesion. Given that the extracted teeth had severe periodontal or dental diseases, it is very unlikely that high loads or fatigue failure occurred. Severely periodontally involved teeth are mobile & forces are thus dissipated by movement / secondary occlusal trauma. The authors need to re-consider their explanation.

Response 6: Thank you for your comment. The patients included in this study had partial edentulism, an inclusion criterion that we failed to mention before, but added in the revised version of the manuscript. Also, we improved the explanation as you recommended.

Point 7. Line 304 The stereomicroscopic examination of NCCL allowed the appreciation of exposed dentin at increased magnification, which highlighted the clinical appearance of hypermineralized dentin The authors must state what is the clinical appearance of hypermineralized dentin. Intra-tubular dentin deposited when odontoblasts retract from stimuli can deposit a hypermineralized dentin but inter-tubular dentin is not altered. The authors must describe what the clinical appearance of hypermineralized dentin is under 75x magnification.

Response 7: Thank you for your suggestion. The stereomicroscopic examination done in this study under up to 75x magnification did not allow the analysis of odontoblasts, intra- and inter-tubular dentine. In this study, the high magnification allowed only the analysis of the appearance of sclerotic dentine in order to establish its score according to data from literature.

Point 8. Furthermore, reference to sclerotic damage on 8. Line 311 is misleading as sclerosis is a physiological protection against insult to the pulp. It is NOT damage.

Response 8: Thank you for your comment. We have changed the term in the revised manuscript to better reflect the meaning.

Point 9. The authors acknowledge the small sample size of 27 teeth but could have provided more detail regarding the origin of the teeth. How many were from males & females. What age groups were represented? For instance, did erosion occur in younger age groups? The aim was partly to associate morphology with etiology but the authors speculated that furrows were caused by tooth brush filaments rather than ask patients what oral hygiene materials/techniques they used. Tooth paste abrasiveness varies & most are safe especially if the RDA value is <45.

Response 9: Thank you for your comment. We added information regarding the age groups represented. We considered that the other information would have overcome the aim of this study, which is to emphasize the role of the stereomicroscopic in examining NCCL. We added another weak point of the study, that it does not offer information regarding the gender of the patients, their diet and oral hygiene habits.

Reviewer 2 Report

Dear Authors, Thank you for the interesting article. Here are some suggestions how to improve It:

1. Line 31, what do you mean by acute trauma

2. LinÄ™ 42+, Please add alternative treatment options, such as natural Polymer and natural toothpastes:

-Paradowska-Stolarz A, Wieckiewicz M, Owczarek A, Wezgowiec J. Natural Polymers for the Maintenance of Oral Health: Review of Recent Advances and Perspectives. Int J Mol Sci. 2021 Sep 25;22(19):10337. doi: 10.3390/ijms221910337.

-Mazur M, Ndokaj A, Bietolini S, Nisii V, DuÅ›-Ilnicka I, Ottolenghi L. Green dentistry: Organic toothpaste formulations. A literature review. Dent Med Probl. 2022 Jul-Sep;59(3):461-474. doi: 10.17219/dmp/146133

3.linÄ™ 56, It would be beneficial to mention the fraktal dimension analysis as a method of analysing the strukture of the cement and root:

- Skośkiewicz-Malinowska K, Mysior M, Rusak A, Kuropka P, Kozakiewicz M, Jurczyszyn K. Application of Texture and Fractal Dimension Analysis to Evaluate Subgingival Cement Surfaces in Terms of Biocompatibility. Materials (Basel). 2021 Oct 7;14(19):5857. doi: 10.3390/ma14195857.

also, add this aspect in line 75

4. Line 126, add Company and country of production

5. As the images of the microscopic view were taken, It is mandatory to add them here. The information on the camera etc should be added

6. In the discussion, Please add DDE Index and SpectroShade evaluation as a quality assasment for the teeth damages

7. In the discussion, you should also pay more attention to non-carious decays caused by clenching, bruxism and TMD in general. 
8. Line 318-320, Please highligh that point - describe It in more detailed way, showin the novelty of YouTube study

9. Format the references (in many, doi is missing) - see the journal’s guidelines

thank you in advance for correcting the article

Author Response

Thank you for your review. 

Response to Reviewer 2 Comments

Thank you for the evaluation and for the recommendations!

Point 1: Line 31, what do you mean by acute trauma

Response 1: When mentioning ,,acute trauma”, we are referring to a single traumatism that can occur during an accident, such as falling from the bicycle. We have addressed this issue and changed the term in the revised manuscript. Thank you!

Point 2: LinÄ™ 42+, Please add alternative treatment options, such as natural Polymer and natural toothpastes.

Response 2: According to the recommendations, we have added that information.

Point 3: linÄ™ 56, It would be beneficial to mention the fraktal dimension analysis as a method of analysing the strukture of the cement and root

Response 3: Thank you for your suggestion. We have addressed the issue.

Point 4: Line 126, add Company and country of production

Response 4: Thank you for your suggestion. We added that information in the revised version of the manuscript.

Point 5: As the images of the microscopic view were taken, It is mandatory to add them here. The information on the camera etc should be added

Response 5: Thank you for your comment. The microscopic images were taken using the NIS-A-AMEAS software, as mentioned now in the revised manuscript.

Point 6: In the discussion, Please add DDE Index and SpectroShade evaluation as a quality assasment for the teeth damages

Response 6: Thank you for your suggestion. We added the specified information to improve the manuscript.

Point 7: In the discussion, you should also pay more attention to non-carious decays caused by clenching, bruxism and TMD in general.

Response 7: Thank you for your suggestion. We have addressed the issue.

Point 8: Line 318-320, Please highligh that point - describe It in more detailed way, showin the novelty of YouTube study

Response 8: Thank you for your suggestion. We have addressed the issue.

Point 9: Format the references (in many, doi is missing) - see the journal’s guidelines

Response 9: Thank you for your comment. We have added doi to all the articles with that indexing in the revised version of the manuscript.

Round 2

Reviewer 2 Report

Thank you for the corrections